# Impact of Acidulants on *Salmonella* and *Escherichia coli* O157:H7 in Water Microcosms Containing Organic Matter

**DOI:** 10.3390/pathogens12101236

**Published:** 2023-10-12

**Authors:** Steven C. Ricke, Elena G. Olson, Christina Ovall, Carl Knueven

**Affiliations:** 1Meat Science and Animal Biologics Discovery Program, Department of Animal and Dairy Sciences, University of Wisconsin, 1933 Observatory Drive, Madison, WI 53706, USA; egolson2@wisc.edu; 2Jones-Hamilton Company, 30354 Tracy Road, Walbridge, OH 43465, USA; covall@jones-hamilton.com (C.O.); knueven.scientific@gmail.com (C.K.)

**Keywords:** *E. coli* O157:H7, Salmonella, sodium bisulfate, lactate, water microcosms, organic matter

## Abstract

As demands for fresh water become more competitive between the processing plant and other consumers of water such as municipalities, interest has grown in recycling or reusing water for food processing. However, recycling the processing water from a poultry plant, for example, represents challenges due to increased organic loads and the presence of bacterial contaminants including foodborne pathogens. The objective in the current study was to evaluate the inactivation of *Salmonella* and *E. coli* O157:H7 using combinations (0.5% and 1%) of sodium bisulfate (SBS) and 1% lactic acid (LA) in water and water with organic matter in the form of horse blood serum (0.3%) with exposure times of 1 min and 5 min. Pathogen reductions after a 5 min exposure time were greater than corresponding reductions after a 1 min exposure time for all acid solutions. The *Salmonella* counts were significantly reduced (i.e., ≥1 log-unit) in all acid solutions after a 5 min exposure time with the combination of LA + SBS acid solutions being more effective than the corresponding 2% LA solutions. None of the acid solutions were effective in reducing the *E. coli* O157:H7 after a 1 min exposure time. The 1% LA + 1% SBS solution was the most effective acid solution against both pathogens and was the only acid solution effective in reducing *E. coli* O157:H7 by at least one log unit after 5 min of exposure.

## 1. Introduction

Water has many applications in the food industry, including irrigation, processing, cooling, heating, and cleaning [1]. Food processing is an essential component of water utilization, with the quality of water supply, generation of substantial effluents, and wastewater treatment representing major expenses [1]. Therefore, water use conservation in the food industry is becoming an increasing environmental and economic concern [1,2]. While water use in a food processing plant would appear incremental at first glance, the cumulative volume of water used in a processing plant can be substantial. For example, in poultry processing, water is used in several stages, including electrical stunning, evisceration, wash steps, scalding, defeathering, deboning, cutup, chilling equipment, and facility sanitation [2,3,4]. Total water use per bird can add up to as much as 26.5 L per 2.3 kg, with evisceration involving the most water use at 7.57 L per bird [2,5]. As demands for freshwater become more competitive between the processing plant and other consumers of water, such as municipalities, interest has grown in recycling or reusing water for food processing [1,2]. Reusing water from a processing plant involves recovery from a processing stage, reconditioning as needed, and subsequent utilization [1,2]. These steps are necessary due to the organic loads in processing water that consist of measurable levels of total nitrogen chemical oxygen demand, along with total suspended solids, and can include adulterants such as oils, proteins, macroparticles, and bacteria [1,2,6]. Therefore, recycling processing water from a poultry plant, for example, represents challenges due to increased organic loads and bacterial contaminants, including foodborne pathogens like *Salmonella* [2,6]. Some bacteria, such as *Salmonella*, fecal coliforms, total coliforms, and *Staphylococcus*, have a zero-tolerance level permitted [2,7].

To decrease organic loads, several physical methods, such as filtration, have been employed to lower chemical contaminants, and chemical interventions have been administered to lower microbial loads [2]. Chemical treatments have traditionally included chlorine, peracetic acid, peroxide, and others such as organic acids, botanicals, phage, and bacteriocins have been suggested as alternative antimicrobials for reusing water [2,8,9]. However, limitations include hazardous handling properties and corrosiveness, while others remain unproven as potential water treatments [2,9]. Inorganic acid sodium bisulfate (SBS) has been suggested as a viable alternative due to its solubility in water and ability to lower pH in moisture without causing off flavors in finished products [2,10]. The use of SBS for direct application on poultry products and vegetables has demonstrated its efficacy against foodborne pathogens such as *Salmonella* Enteritidis and *Listeria* spp. [11,12,13]. The application of SBS for reducing *Salmonella* Typhimurium in reused poultry water has also been examined and shown to be effective in the presence of an organic load [14]. However, the efficacy of SBS in water with known quantities of organic matter remains to be determined. The objective of the current study was to evaluate the inactivation of *Salmonella* and *E. coli* O157:H7 using combinations of SBS (0.5% and 1%) and 1% lactic acid (LA) in water and water with organic matter with exposure times of 1 min and 5 min based on previous work with SBS and reused poultry processing water [14].

## 2. Materials and Methods

### 2.1. Bacterial Strains

The following 5 strains of *Salmonella* were used in this study: *Salmonella* serotype Enteritidis (American Type Culture Collection, ATCC 13076), *Salmonella* serotype Typhimurium (ATCC 14028), *Salmonella* serotype Montevideo (ATCC 8387), *Salmonella* serotype Newport (ATCC 6962), and *Salmonella* serotype Choleraesuis (ATCC 13312). The following strains of *E. coli* O157:H7 were used in this study: ATCC #43894, ATCC #43895, and ATCC #700599. These isolates represent pathogens commonly associated with animals and/or their corresponding meat and poultry products [1,9]. Each bacterial strain was grown separately via at least 2 serial transfers at 35 °C for 24 h in Tryptic Soy broth (TSB). Bacterial cells for each culture were harvested via centrifugation at 10,000× *g* for 10 min and washed twice with Butterfield’s Phosphate Buffer, pH 7.2 (BPB). For each pathogen, each strain was then resuspended, equally combined, and concentrated in BPB to obtain a cell suspension of approximately 5 × 10^9^ CFU/mL. All inoculum suspensions were enumerated on appropriate media (Table 1).

### 2.2. Efficacy Test Procedures

A modification of the AOAC Method 960.09 (Germicidal and Detergent Sanitizing Action of Disinfectants) was used to test the antimicrobial efficacy of the test solutions as follows. (1) For each of the 6 acid test solutions and the 2 assay–control solutions (per Section A), sterile screw cap test tubes (i.e., 8 each) containing 9.9 mL of the subject solution were prepared. The tubes with solutions were equilibrated to room temperature (21–24 °C) before initiating the efficacy tests. (2) For each inoculum, 4 test tubes of each solution prepared as above were inoculated by adding 0.1 mL of the designated inoculum suspension (at 1 × 10^9^ CFU/mL). This yielded an inoculum level of 10^7^ CFU/mL in each tube. The tubes were vortexed immediately after inoculation and prior to each sample aliquot collection. (3) For each inoculum/solution combination, 2 tubes were sampled exactly 1 min (60 s) after inoculation (and exposure from 21 to 24 °C) and 2 more tubes exactly 5 min after inoculation. Tubes were sampled via aseptically transferring a 1.0 mL aliquot to a dilution blank/tube containing 9.0 mL of D-E neutralizing broth to yield an initial dilution of 1:10 (10^−1^).

### 2.3. Enumeration

Sample aliquots from the initial (10^−1^) D-E neutralizing tubes were serially diluted in BPB and enumerated via surface plating on pre-poured plates of Tryptic Soy agar (TSA) agar. TSA plates were incubated for 24 h at 35 °C. Bacterial counts were expressed as colony forming units (CFU) per ml of sample solution and converted to log_10_ transforms. Microbial counts were transformed to log_10_ CFU/mL. The log_10_ and percentage reductions (versus the respective control sample) were calculated for each inoculum/solution/time combination. The pH of the (uninoculated) test solutions was measured. Enumeration data were analyzed using analysis of variance (ANOVA), and pairwise differences were assessed using Tukey HSD with threshold *p*-value cutoff *p* < 0.05.

## 3. Results

The final pH values are shown in Table 1. The non-acidified controls were both near neutral pH at 6.85 and 6.75 with or without horse blood serum (HBS), respectively. Adding 2% LA dropped the pH to 2.34 without HBS and 2.39 in the presence of HBS. The combinations of 1% LA and SBS exhibited slightly lower pH levels at 2.05 and 2.11, respectively, for 0.5% SBS with or without HBS and 2.01 and 1.99, respectively, for 1.0% SBS with or without HBS. In all instances, the addition of HBS appeared to have minimal impact on the pH of the water solution whether acidulants were included or were in the controls without the acidulants. The decrease in pH from the control to the presence of the acidulant was also relatively consistent regardless of whether LA and SBS were combined or were for higher concentrations of LA alone.

Statistical analyses of the results for *Salmonella* inactivation presented as log-unit reductions after 1 and 5 min exposure times in the acid solutions are presented in Table 2. *Salmonella* reductions after a 5 min exposure time were greater than corresponding reductions after a 1 min exposure time for all acid solutions. There was a significant interaction between treatment and time on the *Salmonella* reductions in the current study, and both variables produced a significant effect on *Salmonella* reduction (*p* < 0.05, Table 2).

To assess how the treatments affected *Salmonella* reduction for each time point, the enumeration data were analyzed for each time point separately. For 1 min exposure (Figure 1), the acid solution of 1% LA + 1% SBS (5.66 log_10_ CFU) of recovered *Salmonella* spp. was significantly less than the control (no acidulant) levels of *Salmonella* (7.09 log_10_ CFU) with a ≥1 log-unit reduction. While some reduction was observed to 2% LA and 1% LA + 0.5% SBS (0.44 and 1.01 log-units, respectively), these were not significantly different (*p* > 0.05) from either the control or the 1% LA + 1.0% SBS treatments. Similar results were observed for the treatments in the presence of HBS (Figure 1). The acid solution of 1% LA + 1% SBS (5.65 log_10_ CFU) of recovered *Salmonella* spp. was significantly less than the control (no acidulant) levels of *Salmonella* (7.20 log_10_ CFU) with a ≥1 log-unit reduction. Again, some reduction was observed to 2% LA and 1% LA + 0.5% SBS (0.54 and 1.22 log-units, respectively), but these were not significantly different (*p* > 0.05) from either the control or the 1% LA and 1.0% SBS treatments.

*Salmonella* counts were significantly reduced (i.e., ≥1 log-unit) in all acid solutions after a 5 min exposure time (*p* < 0.05; Figure 2). For the non-HBS containing water samples, the decrease in *Salmonella* populations occurred incrementally as 2.0% LA alone (6.11 log_10_ CFU) significantly reduced log counts greater than 1 log-unit from the control (7.33 log_10_ CFU), over 2 log- units for 1% LA + 0.5% SBS (4.66 log_10_ CFU), and greater than 4 log-units for 1% LA + 1.0% SBS (2.84 log_10_ CFU). *Salmonella* reductions after 5 min in 2% LA solution, 1% LA + 0.5% SBS solution, and 1% LA + 1% SBS solution were 1.22, 2.67, and 4.50 log_10_ units, respectively (equivalent to 93.974%, 99.786%, and 99.997%, respectively; Table 2; Figure 2). For the HBS containing water samples, the decrease in *Salmonella* populations occurred incrementally as 2.0% LA alone (5.88 log_10_ CFU) significantly reduced log counts greater than 1 log-unit from the control (7.18 log_10_ CFU). Both combinations of LA + SBS exhibited similar levels of reduction with greater than 2 log-units for 1% LA + 0.5% SBS (4.51 log_10_ CFU), as well as greater than 2 log-units for 1% LA + 1.0% SBS (4.18 log_10_ CFU). The addition of HBS appeared to reduce the efficacy of the 1% LA + 1.0% SBS but not the 1% LA + 0.5% SBS combination when compared to water samples with no organic matter.

*Escherichia coli* O157:H7 inactivation presented as log-unit reductions after 1 and 5 min exposure times in the acid solutions are provided in Table 3. There was a significant interaction between treatment and time on *E. coli* reductions in the current study as well, as both variables produced a significant effect on *E. coli* reduction (*p* < 0.05, Table 3). To assess how the treatments affected *E. coli* O157:H7 reduction for each time point, the enumeration data were analyzed for each time point separately. Based on ANOVA results, there was a significant effect of treatment and time on the *E. coli* reductions in the current study (*p* < 0.05).

For the 1 min exposure in non-HBS solutions (Figure 3), only the treatment containing 1% LA + 1% SBS produced significantly more *E. coli* O157:H7 reduction (0.3 log-units) compared to the control (7.34 versus 7.04 log_10_ CFU). The addition of HBS eliminated efficacies for all acidulants with no detectable reductions in *E. coli* O157:H7. For the 5 min exposure (Figure 4) to acidulants with no HBS, only the 1% LA + 1% SBS treatment was effective with greater than 1 log-unit reduction from control levels of *E. coli* O157:H7 (7.20 versus 6.17 log_10_ CFU) and 1% LA + 0.5% SBS at 1 min exposure (*p* < 0.05; Figure 3). However, 5 min exposure to treatments of 1% LA + 0.5% SBS and 1% LA + 1% SBS with HBS produced significant *E. coli* O157:H7 reductions (0.61 and 1.19 log-units, respectively) compared to controls (7.48 log_10_ CFU) and were comparable to the 1% LA + 1% SBS (1.03 log-unit reduction) with no HBS treatments (*p* < 0.05, Figure 4). *E. coli* O157:H7 reductions after 5 min with and without organic material in 1% LA + 1% SBS solutions were equivalent to 90.67% and 93.54% reductions in *E. coli* O157:H7, respectively.

## 4. Discussion

The interaction between water quality, microorganisms, and environmental conditions in agricultural use is complex. It can impact various factors, such as detecting foodborne pathogens [15]. As freshwater sources become scarcer, the attractiveness of reusing or recycling water for processing plants in the food industry has become more of a focus for research efforts. Given the zero tolerance for specific pathogens such as *Salmonella*, these pathogens must be eliminated before re-introducing recycled water into the processing plant [2]. The critical issue is developing mitigation methods that are effective in the presence of organic loads of various chemical and biological materials to eliminate pathogens in the water before recycling. While chemicals such as peracetic acid and chlorine have been used to reduce bacterial and pathogen populations on carcasses, they possess limitations that may render them less effective in the presence of sustained levels of organic load [2,9]. For example, peracetic acid generally decomposes in the chiller tank during poultry processing, not reaching the biological poultry wastewater treatment processes [16]. A relatively short half-life of peracetic acid for extended water treatment makes it a less-than-ideal candidate. Other candidates that could serve as acidulants under these conditions would be of interest in achieving pathogen mitigation. The inorganic acid SBS has been examined as an antimicrobial for a variety of meat and vegetable products and has been reviewed for application in reused poultry water [11,12,13,14].

In the current study, SBS was compared and combined with LA in the presence or absence of a pre-determined organic load in the form of HBS. In general, the acid solutions were less effective against *E. coli* O157:H7 than they were against *Salmonella*. There may be differences in tolerance to a low pH between *E. coli* and *Salmonella*, with *E. coli* being resistant to lower pH levels [17,18,19]. In previous research, equivalent reductions in *S*. Typhimurium and *E. coli* O157:H7 were achieved with either SBS or potassium bisulfate (KBS). Still, these were conducted in water-based media, and pH levels were below those measured in the current study [19]. Both studies used cocktails of *Salmonella* spp. and *E. coli* O157:H7, but there could be strain differences in response to acids. In addition, multiple *Salmonella* serovars were used in the cocktail applied in the current study, and *Salmonella* serovars and strains have been shown to range widely in their virulence gene responses to inorganic and organic acids [20]. It is conceivable that acid tolerance is also variable among serovars and strains. When *S*. Typhimurium alone was exposed to 1, 2, or 3% SBS in reused poultry water, a complete reduction below detection was observed within 5 min [14]. Whether this holds for other *Salmonella* serovars is unknown, although the current study with multiple *Salmonella* serovars indicates that most are sufficiently sensitive in the cocktail inoculation. It would be of interest in future studies to examine individual strains and serovars singly to determine if there are differences in response to SBS, LA, and LA + SBS.

Combinations of LA + SBS tended to be more effective than LA alone and for 5 min versus 1 min exposure times. These findings indicate synergistic behavior due to different antimicrobial mechanisms for SBS versus LA. It has been suggested that weak acids such as LA are antimicrobial because they can permeate the cell membrane in both the associated and disassociated forms, disrupting the proton motive force and serving as uncouplers [21]. It has also been suggested that weak acids such as LA are antimicrobial due to the internal accumulation of toxic anions at a low pH [22]. Based on studies with *Salmonella*, *E. coli* O157:H7, and *Listeria monocytogenes*, McDaniels et al. [19] concluded that SBS elicited antimicrobial properties via a multitude of mechanisms involving pH, oxidation of cell components, and osmolarity.

In future studies, genetic and proteomic characterization of *Salmonella* and *E. coli* O157:H7 will be required to determine if there are separate mechanisms for SBS and LA that do not elicit opportunities for cross-protection and if mechanisms are somewhat different between SBS and LA. Understanding the genetic responses of foodborne pathogens to these various antimicrobials could provide the means to design more optimal combinations to achieve further synergistic reductions. In addition, if these antimicrobials are to be applied to reuse water contaminated with a diverse microbial population, it would be essential to determine if the selectivity of more acid-resistant populations occurs. If so, this would require multiple antimicrobials capable of targeting various microorganisms besides foodborne pathogens.

## Figures and Tables

**Figure 1 pathogens-12-01236-f001:**
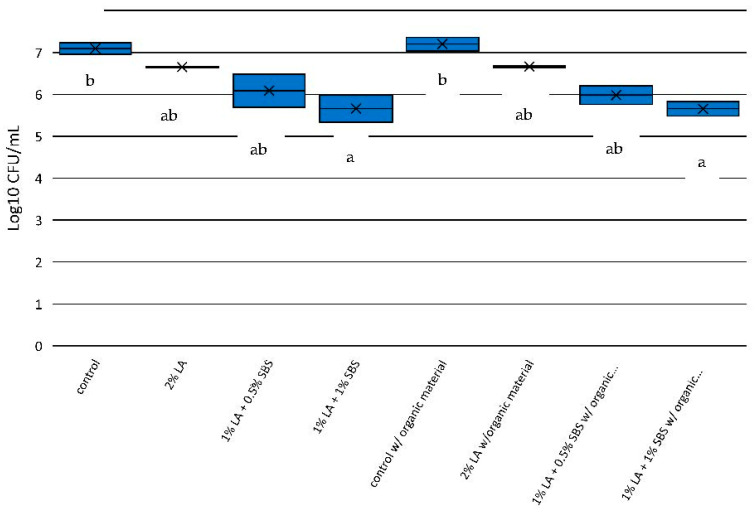
*Salmonella* reduction in aqueous solutions used in the current study after 1 min of exposure to treatments. Significant differences (*p* < 0.05) between treatments are depicted by letters a,b.

**Figure 2 pathogens-12-01236-f002:**
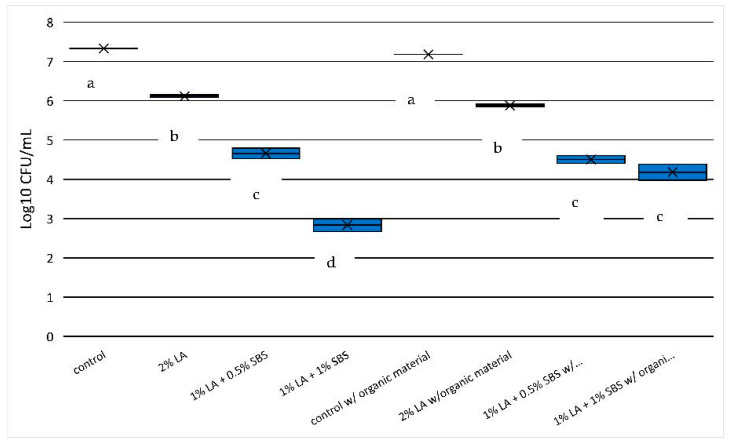
*Salmonella* reduction in aqueous solutions used in the current study after 5 min of exposure to treatments. Significant differences (*p* < 0.05) between treatments are depicted by letters a–d.

**Figure 3 pathogens-12-01236-f003:**
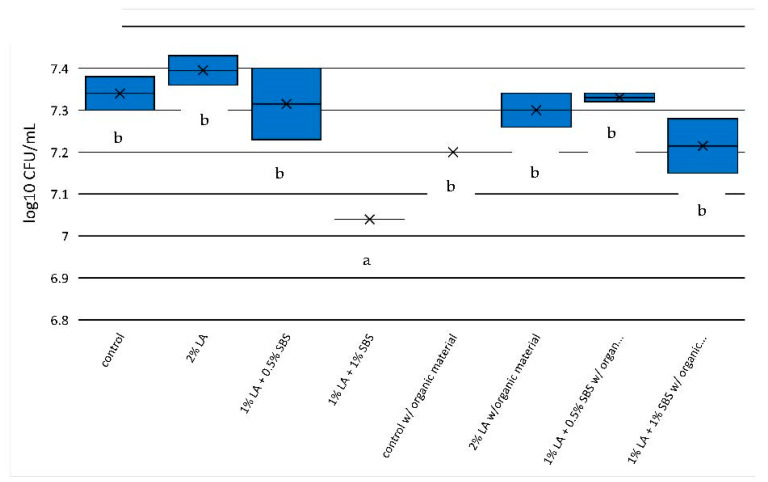
*E. coli* O157:H7 reduction in aqueous solutions used in the current study after 1 min of exposure to treatments. Significant differences (*p* < 0.05) between treatments are depicted by letters a,b.

**Figure 4 pathogens-12-01236-f004:**
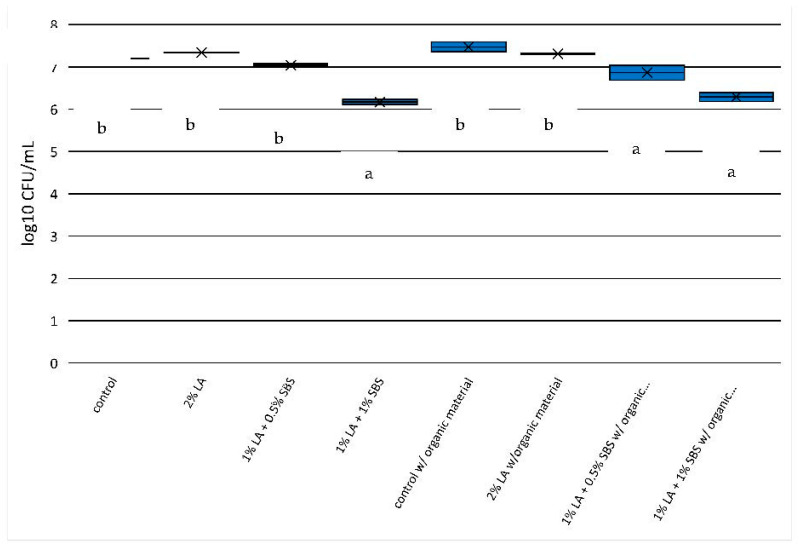
*E. coli* O157:H7 reduction in aqueous solutions used in the current study in 5 min of exposure to treatments. Significant differences (*p* < 0.05) between treatments are depicted by letters a,b.

**Table 1 pathogens-12-01236-t001:** Description of treatments utilized in the current study with the respective pH.

	Treatment Description	pH
1	Control LA (2% lactic acid in DI)	2.34
2	Control LA w/organic matter (2% LA in DI w/0.3% HBS)	2.39
3	1% LA + 0.5% SBS in DI	2.05
4	1% LA + 0.5% SBS in DI w/0.3% HBS	2.11
5	1% LA + 1% SBS in DI	2.01
6	1% LA + 1% SBS in DI w/0.3% HBS	1.99
7	Assay control-1 sterile BPB	6.82
8	Assay control-2 sterile BPB w/0.3% HBS	6.75

Lactic acid solution (LA), deionized water (DI), sodium acid sulfate (SBS), 0.3% horse blood serum (HBS), Butterfield’s phosphate buffer (BPB).

**Table 2 pathogens-12-01236-t002:** ANOVA results for *Salmonella* reduction.

	Degrees of Freedom	Sum Sq	Mean	F Value	Pr (>F)
Treatment (See Figure 1 and Figure 2)	7	32.55	4.65	77.56	3.76 × 10^−11^
Time (See Figure 1 and Figure 2)	1	8.61	8.611	143.63	2.10 × 10^−9^
Treatment/Time	7	6.71	0.959	15.99	3.96 × 10^−6^
Residuals	16	0.96	0.06		

**Table 3 pathogens-12-01236-t003:** ANOVA results for *E. coli* O157:H7 reduction.

	Degrees of Freedom	Sum Sq	Mean Sq	F Value	Pr (>F)
Treatment (See Figure 3 and Figure 4)	7	2.2764	0.3252	33.69	2.04 × 10^−8^
Time (See Figure 3 and Figure 4)	1	0.7473	0.7473	77.41	1.58 × 10^−7^
Treatment/Time	7	1.2555	0.1794	18.58	1.43 × 10^−6^
Residuals	16	0.1544	0.0097		

## Data Availability

Data is privately held but can be made available upon reasonable request.

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
