# Peer review of "Impact of Acidulants on Salmonella and Escherichia coli O157:H7 in Water Microcosms Containing Organic Matter"

_pathogens, 2023, doi:10.3390/pathogens12101236_

Round 1

Reviewer 1 Report

The manuscript entitled “Impact of Acidulants on Salmonella and Escherichia coli O157:H7 in Water Microcosms Containing Organic Matter” shows the germicidal effect of two different types of sanitizing agents, sodium bisulfate (SBS) and lactic Acid (LA), applied separated or combined during 1 or 5 min against microorganisms. In general, the authors found that the combination of both solutions during 5 min is the most effective. The data presented in this manuscript rise important information about the proper way to reuse water for poultry plant system.

For this referee the manuscript brings very conclusive data, however, as the authors themselves pointed out in the text, there no information of how the sanitizers could be affecting the different serotypes of Salmonella or the strains of E. coli. It is very known that the manuscript focuses on study the effect of these sanitizers in water containing organic matter, but why the authors did not test the sanitizers in each one of the microorganisms found in the used Salmonella and E. coli cocktail? If it is already known that different types of serotypes or stains could react in a different way to each sanitizer, why did the authors preferred to postpone these results? 

As a major revision:

1- The analysis of sanitizers should be done with the microorganisms separately not just in the cocktail. In that way, the data will be more complete given support to the conclusion section.

2- What is the cytotoxic effect of SBS? Was it verified whether the combination of LA and SBS at the highest concentration used in the present work is toxic? If this data has already been published, could the authors include this information in the discussion session? There are some studies that show the cytotoxic potential of SBS in animal cells (doi: 10.1016/j.etap.2015.08.013; 10.1016/j.etap.201 4.01.019; 10.1016/j.etap.2016.02.004).

As minors revisions:

1- The table 2 (page 3, line 126) is missing some explanation: What is Df? Where are the indication about what kind, concentration and time of sanitizers exposition the data refer? Please this information should be added in someplace in the results section.

2- In all figures (1 – 4), there is no explanation what the letters a, b and c are related? Is there related to the P value? If it is the case, what means a, b and c, more or less significant? This should be explained in the legends or in methods section.

3- The figure 1, there is no c letter, but is mentioned in the legend, please correct.

4- In the discussion section, line 210, the authors mention that in this study their just compared the effect of SBS with LA, but besides the comparison its was showed a combined effect, so please add in the sentence: In the current study, SBS was compared and combined with LA in the presence…

Reviewer 2 Report

The study and results presented in this manuscript are helpful and worthy of publication consideration. The presented data and results provide insights for further experimental designs to examine chemical disinfectants/agents for pathogens inactivation in recycled wastewater. Following are some recommendations for improvement:

Line 54-55: The statement should be supported by the reference source(s) citation.

Line 65-66: Describe the specific reasoning for 1 min and 5 min exposure times for the inactivation experiments.

Line 75: Reference source should be cited.

The figures captions should be clarified to better explains the notations for statistical significance, indicate that the organic matter label corresponds with HBS in the solution, and specify that the x inside the box plots is the mean for how many replicates. It would also be helpful to separate the regions, by a dotted line, for the different experiments and corresponding control groups.

The document should be checked for spelling and grammar. For example, line 22, Abstract, solutions was, should be solutions were.

Reviewer 3 Report

Dear Authors

Thanks for manuscript with title: Impact of Acidulants on Salmonella and Escherichia coli O157:H7 in Water Microcosms Containing Organic Mater, the review of the aforementioned manuscript has been finished and there are some points about it which you could find at the attached file.

Best Regards

Round 2

Reviewer 1 Report

Now, for this referee, the manuscript is suitable for publication.